# A Study on the Equity Dilemma and Reform Strategies of Drug Reimbursement in China’s Medical Insurance System

**DOI:** 10.3390/healthcare13202646

**Published:** 2025-10-21

**Authors:** Minghao Yang, Yumeng Zhang, Qiang Su, Yuanhao Sui, Lihua Sun

**Affiliations:** Department of Pharmacy Administration, School of Business Administration, Shenyang Pharmaceutical University, 103 Wenhua Road, Shenyang 110016, China; ymh9077@163.com (M.Y.); ymzhang202204@163.com (Y.Z.); 13540168209@163.com (Q.S.); 15564199762@163.com (Y.S.)

**Keywords:** medical insurance reimbursement, equity, method of payment

## Abstract

**Background:** The continuous expansion of the National Reimbursement Drug List has led to an increasing cost disparity among alternative drugs for the same indications. Under the current proportional reimbursement mechanism, choosing higher-cost treatments often results in higher compensation. Given the lack of empirical evidence on whether income affects the medication choices of insured individuals in the Chinese context, this study aims to evaluate the impact of income levels on drug selection, providing a basis for optimizing the medical insurance reimbursement policy. **Methods:** This study extracts data from hospitalized patients enrolled in basic medical insurance from the China Health and Retirement Longitudinal Study (CHARLS) database and preprocesses it in Excel. Subsequently, SPSS is used to conduct descriptive statistics, difference analysis, correlation analysis, and regression analysis on the processed data to explore the impact of income levels on drug selection. **Results:** After controlling for length of hospitalization and hospitalization costs, the regression coefficient for urban employee basic medical insurance participants is *β* = 0.505 (*p* < 0.01), and the regression coefficient for new rural cooperative medical insurance participants is *β* = 0.195 (*p* < 0.01). This means that, regardless of whether participants are enrolled in urban employee basic medical insurance or new rural cooperative medical insurance, an increase in income will lead to higher hospitalization drug costs. **Conclusions:** Compared to low-income insured individuals, high-income participants in the basic medical insurance are more likely to choose higher-cost drugs among alternatives, which leads to unfair reimbursement under the current proportional reimbursement system.

## 1. Introduction

In modern medical systems, almost all medical activities, including inpatient treatment, surgical procedures, and long-term management of chronic diseases, involve pharmaceuticals. Drugs play a pivotal role throughout the entire process of disease diagnosis and treatment [1]. According to the 2024 National Medical Security Development Statistical Bulletin, total expenditures for basic medical insurance (including maternity insurance) reached 2.967 trillion RMB, with the amount spent on drugs procured through provincial centralized procurement platforms within the insurance catalog amounting to 922.5 billion RMB [2]. The insurance catalog is divided into two categories: Category A drugs are reimbursed in accordance with the prescribed ratios of each region. Unlike Category A drugs, Category B drugs are subject to coinsurance: patients first pay a percentage share out of pocket, and the remaining eligible amount is reimbursed at the prescribed regional rate [3]. Based on these figures, it is estimated that the reimbursement for drugs accounts for about one-quarter of the total medical insurance fund expenditure. Given that some designated medical institutions do not procure drugs from the provincial procurement platform [4], the actual reimbursement proportion for drugs is likely higher, indicating that drugs occupy a significant share of medical insurance fund expenditures.

With advancements in technology [5] and the expansion of medical insurance fund size, the insurance catalog has continually expanded [6], encompassing an increasingly diverse range of drug types and categories [7]. As a result, the cost disparities among substitute drugs for the same indication have gradually increased, creating a clear cost gradient. As more innovative drugs have been included in the insurance catalog in recent years [8], these disparities have been further exacerbated. China’s current medical insurance system is based on the generic name of the drug, with differentiated reimbursement rates set according to drug categories (Category A and Category B) [3]. This system implies that, for specific indications, if a patient selects a higher-cost drug from the list of substitutes, the out-of-pocket cost will increase after reimbursement. Still, the reimbursement amount will also rise correspondingly. High-income patients, possessing a greater ability to pay out-of-pocket expenses, may be more inclined to choose higher-cost drugs, resulting in higher absolute reimbursement amounts, thus raising concerns about fairness—where reimbursement amounts are not solely determined by a patient’s actual medical needs but are also influenced by non-medical factors, such as income [9,10]. In this context, it is crucial to explore whether high-income patients are more likely to select higher-cost drugs from the list of substitutes for the same indication.

Existing studies, such as one in Belgium, found that patients with lower socioeconomic status are more likely to choose lower-cost generic drugs, with their drug expenses significantly lower than those of patients with higher socioeconomic status [11]. This conclusion highlights the key role of income level in drug selection. However, systematic studies on the relationship between income and drug selection among Chinese patients remain scarce, especially in light of China’s unique health insurance policies, drug varieties, reimbursement ratios, and patient payment capabilities. Whether similar conclusions can be drawn under these conditions still requires further exploration. Based on this, this study aims to investigate whether, in the unique socioeconomic and health insurance context of China, high-income basic medical insurance enrollees are more likely to select higher-cost drugs among interchangeable options compared to low-income enrollees. The goal is to provide empirical evidence for optimizing drug reimbursement policies in China and to offer decision-making support for related policy formulation.

## 2. Materials and Methods

### 2.1. Data Sources

The data for this study comes from the China Health and Retirement Longitudinal Study (CHARLS) [12], led by Peking University. The survey covers samples from 28 provinces (including autonomous regions and municipalities directly under the central government) and collects information on various aspects such as individual living conditions, health and medical utilization, health insurance enrollment and coverage, and household income. These aspects meet the analytical requirements of this study.

Due to the lack of key variables required for this study in other publicly available Chinese databases, and because CHARLS removed relevant modules after 2015, this study selects data from the 2015 CHARLS survey as the research sample. This survey was launched in 2015 and concluded in 2017, so the analysis window for this study is defined as 2015–2017. The original dataset contains 697 records, and after data cleaning and preprocessing, 480 valid samples remain for analysis. Although the conclusions of this study are based on 2017 data, income disparities among patients still exist, and the cost differences between alternative drugs for the same indication have gradually increased, with limited reimbursement rates. Therefore, economic status remains an important factor in patients’ decision-making when faced with drugs of varying treatment costs. The conclusions drawn based on the data from 2017 remain largely applicable. However, it should be noted that patients’ preferences for drug selection may be influenced by various factors, such as advancements in medical technology and the increase in health awareness, and these factors were not fully taken into account in this study. As China’s real-world drug data system continues to improve, if updated relevant data becomes available in the future, using this new data for validation could further enhance the applicability and accuracy of the research findings.

### 2.2. Variable Selection and Definition

Based on the research objectives, literature review [13,14,15,16], and data availability, the independent variable in this study is set as per capita income, representing the average income of each household member. This variable is derived from the “Work, Retirement, and Pension” and “Income, Expenditure, and Assets” modules in the CHARLS questionnaire. The dependent variable is set as inpatient drug expenses, which refers to the total expenditure on drugs incurred during a patient’s hospitalization, including both out-of-pocket payments and insurance reimbursements. The related data is sourced from the “Medical and Insurance” module of the CHARLS questionnaire. To control for the potential interference of disease severity and types of diseases on drug costs, and to ensure that the drug expenses accurately reflect the cost variation caused by differences in drug selection under the same indication, the study initially includes age, gender, length of hospitalization, and total hospitalization expenses as control variables. Based on this, difference analysis will be conducted to further filter and finalize the control variables, enhancing the scientific rigor and reliability of the empirical results.

### 2.3. Research Design and Methods

This study first extracted data on hospitalized patients participating in basic medical insurance from the CHARLS database using STATA and performed data preparation in Excel. The data preparation process included: removing missing values, excluding data where inpatient drug expenses are greater than or equal to total hospitalization expenses, and categorizing samples based on insurance type (Urban Employee Basic Medical Insurance, Urban Resident Basic Medical Insurance, and New Rural Cooperative Medical Scheme) to ensure the validity and reliability of the analysis results. Due to a small sample size in the Urban Resident Basic Medical Insurance group, which does not meet the requirements for subsequent statistical analysis, this group is excluded from the study. Subsequently, SPSS Version 17 was used to perform descriptive statistics on the research variables, including mean values and range calculations. Based on this, difference analysis will be conducted to select and finalize the control variables, followed by correlation analysis and regression analysis to explore the correlation between per capita income and inpatient drug expenses and to assess the impact of this relationship.

## 3. Results

### 3.1. Descriptive Statistics of the Sample

In the sample of patients enrolled in Urban Employee Basic Medical Insurance, the ages ranged from 42 to 84 years, with an average age of 64. The length of hospitalization ranged from 0 to 150 days, with an average duration of 14.5 days. Hospitalization costs ranged from 1000 to 150,000 RMB, with an average hospitalization cost of 18,272 RMB. Inpatient drug expenses ranged from 100 to 96,000 RMB, with an average of 7987.5 RMB. Per capita income ranged from 0 to 93,000 RMB, with an average income of 11,120.274 RMB. The detailed results are shown in Table 1.

For patients enrolled in the New Rural Cooperative Medical Scheme, the age ranged from 21 to 90 years, with an average age of 61. The length of hospitalization ranged from 0 to 60 days, with an average duration of 11.4 days. Hospitalization costs ranged from 80 to 140,000 RMB, with an average of 9495.8 RMB. Inpatient drug expenses ranged from 6 to 83,000 RMB, with an average of 3516 RMB. Per capita income ranged from 0 to 71,196 RMB, averaging 6518 RMB. The detailed results are shown in Table 1.

### 3.2. Analysis of the Differences in Inpatient Drug Expenses by Gender, Age, Length of Stay, and Hospitalization Cost

Based on the described analysis, those with an age less than or equal to the average were categorized as the low group and those with an age greater than the average as the high group. For length of stay and hospitalization costs, those with values below the average were categorized as the low group and those with values above the average as the high group in order to study the differences in inpatient medication costs. Additionally, tests of skewness and kurtosis for inpatient medication expenses showed that the inpatient medication expenses for urban employees’ basic medical insurance follow a normal distribution (kurtosis = 8.341, skewness = 2.458), while the inpatient medication expenses for the New Rural Cooperative Medical Scheme do not follow a normal distribution (kurtosis = 6.108, skewness = 52.343). The testing criteria are that the absolute value of kurtosis should be less than 10 and the absolute value of skewness should be less than 3. Therefore, an independent samples *t*-test was used to examine the differences in inpatient medication expenses based on gender, age, length of stay, and hospitalization costs for participants in urban employees’ basic medical insurance. Non-parametric tests were used to study the differences in inpatient medication expenses based on gender, age, length of hospital stay, and hospitalization costs for participants in the New Rural Cooperative Medical Scheme. The detailed analysis results are as follows.

The results showed that gender and age did not exhibit significant differences in inpatient drug expenses (*p* > 0.05), indicating that there was no difference in drug expenses based on gender or age. However, length of stay and hospitalization costs showed significant differences (*p* < 0.05), indicating that these variables had a significant impact on inpatient drug expenses. Specifically, length of stay had a significant effect at the 0.01 level (t = −2.942, *p* = 0.005), with the average drug expenses for patients with shorter stays (4345.51 RMB) being significantly lower than those with longer stays (14,792.13 RMB). Similarly, hospitalization costs also showed a significant effect at the 0.01 level (t = −3.974, *p* = 0.000), with patients with lower hospitalization costs (3227.56 RMB) having significantly lower drug expenses compared to those with higher hospitalization costs (21,757.14 RMB), as shown in Table 2.

Gender does not show a significant effect on inpatient medication expenses (*p* > 0.05), which means there is no difference in inpatient medication expenses between different gender samples. Age, length of stay, and hospitalization costs show significant effects on inpatient medication expenses (*p* < 0.05), indicating that these factors have differences in inpatient medication expenses. Specifically, the median for the younger age group (2000.000 RMB) is significantly higher than that of the older age group (1000.000 RMB); the median for the shorter stay group (1000.000 RMB) is significantly lower than that for the longer stay group (3000.000 RMB); and the median for the lower hospitalization cost group (1000.000 RMB) is significantly lower than that for the higher hospitalization cost group (5200.000 RMB). The detailed results are shown in Table 3.

### 3.3. Correlation Analysis of Inpatient Drug Expenses

The Pearson correlation coefficient between per capita income and inpatient drug expenses for patients enrolled in Urban Employee Basic Medical Insurance was 0.523, while for patients enrolled in the New Rural Cooperative Medical Scheme, the correlation coefficient was 0.538. Both were statistically significant at the 0.01 level, indicating a significant positive correlation between per capita income and inpatient drug expenses in both groups. Therefore, further regression analysis was conducted for both groups.

### 3.4. Linear Regression Analysis of Inpatient Drug Expenses

The per capita income of insured Urban Employee Basic Medical Insurance patients was used as the independent variable, with hospitalization duration and hospitalization costs as control variables and hospitalization drug costs as the dependent variable for linear regression analysis. The per capita income of insured New Rural Cooperative Medical Insurance patients was used as the independent variable, with age, hospitalization duration, and hospitalization costs as control variables and hospitalization drug costs as the dependent variable for linear regression analysis. The detailed results are shown in Table 4. The analysis results of insured Urban Employee Basic Medical Insurance patients are as follows. The F-test indicated that the overall model was significant (F = 49.247, *p* = 0.000 < 0.05), suggesting that at least one of the variables—length of stay, hospitalization costs, or per capita income—has a significant impact on inpatient drug expenses. The variance inflation factors (VIFs) were all less than 5, and tolerance values were greater than 0.2, indicating that no multicollinearity issues exist. The Durbin–Watson statistic was close to 2, suggesting that there is no autocorrelation in the residuals, and the model fits well. The regression coefficient for per capita income was 0.505 (t = 8.087, *p* = 0.000 < 0.01), indicating that higher per capita income is significantly associated with increased inpatient drug expenses. The analysis results of insured New Rural Cooperative Medical Insurance patients are as follows. The F-test indicated that the overall model was significant (F = 162.212, *p* = 0.000 < 0.05), suggesting that at least one of the variables—length of stay, hospitalization costs, or per capita income—has a significant impact on inpatient drug expenses. The variance inflation factors (VIFs) were all less than 5, and tolerance values were greater than 0.2, indicating that no multicollinearity issues exist. The Durbin–Watson statistic was close to 2, suggesting that there is no autocorrelation in the residuals, and the model fits well. The regression coefficient for per capita income was 0.195 (t = 8.994, *p* = 0.000 < 0.01), indicating that higher per capita income is significantly associated with increased inpatient drug expenses.

## 4. Discussion

China’s basic medical insurance system adopts a payment model based on generic drug names and sets differentiated reimbursement rates according to drug categories. In the early stages, this approach balanced operability and fairness, meeting the needs of the initial medical insurance management system. However, as the insurance catalog expands, the cost differences for interchangeable drugs under the same indication continue to widen. The empirical results show that after indirectly controlling for confounding factors such as disease type and severity, income level is positively correlated with drug expenses, which largely reflects the preference of high-income patients for higher-priced options within the interchangeable drug set. The current generic drug-based proportional reimbursement mechanism has failed to effectively regulate these cost differences, leading to significant fluctuations in actual fund payments based on individual drug selection costs. Patients who choose more expensive alternatives pay more out of pocket but receive a higher absolute reimbursement, objectively leading to a greater fund compensation for high-cost choices, i.e., a “reverse subsidy,” which deviates from the original intention of fairness in the medical insurance system.

In this context, China’s medical reimbursement system could draw on Germany’s reference pricing system [17,18,19,20] to regulate the payment amounts for substitute drugs under the same indication in the healthcare insurance catalog, addressing the fairness challenges currently faced by the reimbursement system. The core of this model is as follows: First, based on factors such as the income level of insured individuals, market prices of drugs, and market shares, a standard drug (including its packaging specifications) is selected. On this basis, the reimbursement amount for the standard drug is set according to the affordability of the medical fund. Other drugs will be referenced against the standard drug, with their reimbursement amounts determined based on differences in specifications, dosages, treatment durations, and average daily doses. The significant advantage of this payment method is that it breaks the direct link between drug costs and reimbursement amounts, thus avoiding unfairness at the institutional level.

Currently, China’s drug approval and healthcare insurance coding systems still follow the “generic name” (i.e., chemical molecule) management model, with generic names serving as the smallest unit of management from drug registration, the insurance catalog, to the settlement system [21]. This system is not yet fully aligned with the new reimbursement policy framework. Additionally, some drugs that previously enjoyed higher reimbursement rates under the old system may experience reduced reimbursement amounts under the new model, which could lead to dissatisfaction among patients, thus complicating the implementation of the policy. In this context, a “dual-track progressive evolution” strategy could serve as an effective response. Specifically, newly added innovative drugs to the insurance catalog could be used as a pilot for the new reimbursement policy, while retaining the original reimbursement model for older drugs [22]. This “new drugs with new models, old drugs with old models” strategy would allow for the gradual accumulation of practical experience, strengthening policy formulation, technical support, and management capabilities, while also providing a transition period for patients to adjust to the new policy. This approach would lay a solid institutional foundation for future widespread implementation. On one hand, the relatively small number of innovative drugs and their concentrated patient population make them ideal for piloting, offering clear boundaries and manageable risks. On the other hand, the high prices of these drugs, if the current proportional reimbursement model is maintained, would not only exacerbate the imbalance in benefits across income groups but also significantly increase the financial pressure on the healthcare fund.

For single-indication innovative drugs that are newly added to the catalog, if a clinically equivalent substitute exists, the reimbursement amount will be calculated using the method described above. However, if no equivalent substitutes are available in the insurance catalog, the current proportional reimbursement model may continue temporarily. For innovative drugs with multiple indications, reimbursement rules should be developed based on the availability of substitute drugs for each indication in the medical insurance catalog. When implementing the new reimbursement model, the system design must be scientifically sound, and robust regulatory mechanisms must be established to ensure the fair, just, and efficient implementation of the policy. To prevent certain medical institutions or individuals from manipulating diagnoses, fabricating conditions, or maliciously selecting indications with higher reimbursement amounts to exploit the system, medical authorities should establish a comprehensive spot-checking mechanism covering medical institutions, designated pharmacies, doctors, and patients. Random inspections, targeted monitoring, and data comparison should be used to identify abnormal claims promptly. Once malicious “diagnosis adjustments” or fabricated indications are confirmed, strict penalties should be enforced in accordance with regulations, creating a strong deterrent effect through the “punish one case, deter many” approach. In light of China’s current dual-channel policy for innovative drugs (i.e., drugs can be purchased through both medical institutions and designated retail pharmacies), pharmacies [23], as the “frontline” of medical reimbursement, must assume the role of gatekeepers. On one hand, designated pharmacies should strictly enforce the prescription drug purchase management system to ensure that patients only buy drugs based on genuine and standardized prescriptions and strictly prohibit behaviors such as “selling drugs without prescriptions” or “prescription fraud” On the other hand, regulatory authorities should strengthen inspections, audits, and penalties for retail pharmacies, implementing the “who dispenses, who is responsible” accountability system to prevent individuals from illegally purchasing drugs under the guise of higher-reimbursement indications by fabricating prescriptions or using others’ diagnostic information.

To better and more quickly implement the regulation of reimbursement amounts for substitute drugs under the same indication in the future, the author recommends continued reform and capacity building in the following key areas. First, the construction of a medical reimbursement infrastructure centered around drug indications should be accelerated, improving the standardized mapping system between “drug–indication–diagnosis–payment” to provide a data foundation and technical support for differentiated reimbursement. Second, efforts should be made to promote the deep integration of medical insurance information systems with hospital diagnostic systems, enabling real-time linkage and review of prescription, diagnosis, and payment data, thereby enhancing the precision and controllability of payment management. At the same time, the payment amounts for various alternative drugs for different indications under medical insurance should be dynamically adjusted in light of dimensions such as per capita income, medical insurance funds, disease burden, and drug prices. Moreover, the capacity for medical fund auditing and supervision should be strengthened, with regular operation of intelligent auditing, behavioral recognition, and accountability mechanisms to prevent system abuse. Finally, training and guidance for doctors, pharmacists, and patients should be strengthened to improve their understanding and cooperation with the new reimbursement model, creating a favorable environment for policy implementation. In addition, there is another issue worth further exploration: whether high-income insured patients, who tend to choose higher-cost medications among substitute medications, experience better health outcomes. While high-income patients are inclined to choose more expensive drugs, it remains to be verified through more in-depth empirical research whether this choice truly leads to significant health improvements. Future research should focus on the health changes of high-income groups after selecting higher-cost medications and the actual impact of these choices on treatment outcomes, thereby providing more scientific evidence for the optimization of healthcare policies. This study has certain limitations. First, the data used in this study comes from the 2015–2017 CHARLS survey. Given that patient behavior and preferences may change over time, caution is needed when extrapolating the conclusions to the present. Second, due to the lack of generic and brand-name drug information, we could only represent the choice of interchangeable drugs for the same indication using “drug expenses.” Third, the control variables are limited by data availability; this study could only include age, gender, length of hospitalization, and hospitalization costs, which partially substitute for controlling confounding factors such as disease type and severity.

## 5. Conclusions

After indirectly controlling for confounding factors such as disease type and severity, the results show that high-income basic medical insurance enrollees are more likely to choose higher-cost drugs compared to low-income enrollees, which largely reflects their preference for higher-priced options within the set of interchangeable drugs. With the increasing richness of the medical insurance catalog and the current proportional payment framework, the corresponding reimbursement amounts are also relatively higher. To enhance the fairness of fund allocation, it is recommended to regulate the payment amounts for interchangeable drugs under the same indication. However, there is currently a lack of underlying governance and data systems to support the operation of this model. Therefore, a feasible approach could be to use newly included innovative drugs in the medical insurance catalog as an entry point, initiating pilot projects for new payment methods, while simultaneously promoting the joint development of data standards and information systems centered on indications between medical institutions and medical insurance management departments, laying the foundation for a more flexible, precise, and sustainable medical insurance payment mechanism.

## Figures and Tables

**Table 1 healthcare-13-02646-t001:** Descriptive Statistics of the Sample.

Variable	Urban Employee Basic Medical Insurance *N* = 109			New Rural Cooperative Medical Scheme *N* = 307		
	Min	Max	Mean	Min	Max	Mean
Age	42.000	84.000	63.761	21.000	90.000	60.774
Length of Stay	0.000	150.000	14.500	0.000	60.000	11.407
Hospitalization Cost	1000.000	150,000.000	18,271.651	80.000	140,000.000	9495.803
Inpatient Drug Expenses	100.000	96,000.000	7987.450	6.000	83,000.000	3516.011
Per Capita Income	0.000	93,000.000	11,120.274	0.000	71,196.000	6517.598

**Table 2 healthcare-13-02646-t002:** Analysis of Differences in Inpatient Drug Expenses by Gender, Age, Length of Stay, and Hospitalization Costs (Urban Employee Basic Medical Insurance).

Variable	Group (*n*)	Inpatient Drug Expenses (Mean ± SD)	t	*p*
Gender	Male (*n* = 58)	7059.97 ± 9990.32	−0.69	0.492
	Female (*n* = 51)	9042.24 ± 19,117.12		
Age	Low Age Group (*n* = 56)	10,151.95 ± 18,953.22	1.596	0.115
	High Age Group (*n* = 53)	5700.43 ± 8516.82		
Length of Stay	Short Stay (*n* = 71)	4345.51 ± 8509.50	−2.942	0.005
	Long Stay (*n* = 38)	14,792.13 ± 20,988.25		
Hospitalization Costs	Low Cost (*n* = 81)	3227.56 ± 2632.53	−3.974	0.000
	High Cost (*n* = 28)	21,757.14 ± 24,623.30		

**Table 3 healthcare-13-02646-t003:** Analysis of Differences in Inpatient Drug Expenses by Gender, Age, Length of Stay, and Hospitalization Costs (New Rural Cooperative Medical Scheme).

Variable	Group (*n*)	Inpatient Drug Expenses (Median)	t	*p*
Gender	Male (*n* = 179)	1700	−1.521	0.54
	Female (*n* = 192)	1227		
Age	Low Age Group (*n* = 176)	2000	−2.19	0.029
	High Age Group (*n* = 195)	1000		
Length of Stay	Short Stay (*n* = 247)	1000	−6.892	0.000
	Long Stay (*n* = 124)	3000		
Hospitalization Costs	Low Cost (*n* = 268)	1000	−8.898	0.000
	High Cost (*n* = 103)	5200		

**Table 4 healthcare-13-02646-t004:** Linear Regression Analysis.

Variable	Urban Employee Basic Medical Insurance			New Rural Cooperative Medical Scheme		
	B	VIF	Tolerance	B	VIF	Tolerance
Constant	−25,661.864 **	-	-	697.542	-	-
Age				−22.909	1.002	0.998
Length of Stay	6090.889 **	1.114	0.897	−4.286	1.158	0.864
Hospitalization Costs	15,765.731 **	1.115	0.896	0.315 **	1.312	0.762
Per Capita Income	0.505 **	1.002	0.998	0.195 **	1.157	0.864
R^2^	0.585			0.639		
Adjusted R^2^	0.573			0.635		
F	F (3105) = 49.247, *p* = 0.000			F (4366) = 162.212, *p* = 0.000		
Durbin-Watson	1.421			2.038		

Note: ** *p* < 0.01.

## Data Availability

The original data presented in the study are openly available in China Health and Retirement Longitudinal Study, CHARLS at https://charls.pku.edu.cn/.

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
