# Peer review of "A Study on the Equity Dilemma and Reform Strategies of Drug Reimbursement in China’s Medical Insurance System"

_healthcare, 2025, doi:10.3390/healthcare13202646_

Round 1

Reviewer 1 Report

Comments and Suggestions for Authors

The manuscript has many good elements and practical implications. On the other hand, there are several aspects of manuscript that create some concern and require more attention.

1. The Background sub-section in the abstract is too extensive and includes the results of other studies. It would be good to adapt this part by focusing on the essence with the obligatory reduction of the amount of text.

2. At the end of the Introduction section, it is necessary to clearly define the purpose and goal of the research. Also, it is necessary to clearly define what new the existing study brings. The results of the study from Belgium are very similar to the results of the study presented in the manuscript. What's new? Were different results expected so the study was replicated? The results are extremely expected - that those who pay higher insurance premiums expect the use of more expensive drugs. Are different results expected? If so - why? If not - what is the novelty of the study? It is necessary to strengthen the purpose and originality of the study.

3. In the description of the data used in the study, it was stated that data from the CHARLS database was used, starting from 2015, but it was not stated what the final year was. Essentially, the time period for which the data was used should be specified.

4. What are the limitations of the study? It is necessary to clearly position all limitations of the study.

5. The Conslusion section is completely missing. It is recommended that this section be included in the structure of the manuscript and that its content be equivalent to the usual practice for this type of manuscript.

Reviewer 2 Report

Comments and Suggestions for Authors

It is an interesting study and provides some evidence for insurance and drug pricing design. However, I do feel that the methodology is not correctly implemented. I suggest authors only keep the descriptive results and drop linear regression. This is a very complicated research question, which needs a very deliciated statistical design. 

Major comments:

  1. I understand that data after 2015 is not available, but the introduction indicates, "With advancements in technology and the expansion of medical insurance fund size, the insurance catalog has continually expanded, encompassing an increasingly diverse range of drug types and categories." Does using 2015 data still fit this provided framework?

  2. Considering the small sample size and large variations, would comparing means still make sense? Please compare medians and IQR for more precise estimation.

  3. You cannot perform linear regression on "Inpatient drug expense" with "length of stay, costs, and income." This needs to be a structural model because patients will spend as much as they need for healthcare (inelastic demand), but low income patients cannot spend more money for higher price drug. For example, lower-income patients cannot spend more even if they want to, which could bias your results towards to high income patients spend more. In other words, even if lower-income patients would want to spend more, they cannot to do so since they are income-restricted. So, in your regression, you will only observe that patients with low income spent less, and it is not possible for them to spend more money. This violates the linear regression assumptions unless you can argue that it is possible for patients to borrow money and receive high-quality care in China.

  4. Even if design was valid, I am not sure how you reached the conclusion about drug price and reverse subsidy. The only results you are showing are that higher income leads to more drug expense. This result and your conclusion has a large gap at this moment. Moreover, you did not control for the severity of the diseases and many other confounders in your regression. 

Minor comments:

  1. The abstract background section was too long, and not enough results were shared.

  2. Line 46 states, "pay a certain percentage upfront, with the remaining portion reimbursed proportionally." Do you mean copay? I understand it might not be a term used in China, but please update those terms globally to align with English literature.

Round 2

Reviewer 1 Report

Comments and Suggestions for Authors

The authors made changes related to the suggestions given for revising the manuscript. Most of them are well addressed and do not leave additional dilemmas. However, one fact requires additional attention. The authors stated that the study is based on data from the period 2015-2017. This is a significant limitation and therefore requires additional argumentation. First, in the Data Sources section, it is necessary to explain the significance of the data from 2015 for 2025. Are the user preferences the same after 10 years? What could have influenced the change? What could have contributed to the fact that user preferences have not changed in the past 10 years? What are the weaknesses of such an approach? Second, it seems that the part of the text dealing with limitations is better positioned at the end of the Discussion section than at the end of the Conclusion section. Third, in the limitation that refers to the years from which the data were used in the study, it is necessary to state the real limitations in a fair way and thus strengthen the credibility of the study.

Reviewer 2 Report

Comments and Suggestions for Authors

Authors have fully addressed my comments. Congrats to authors for a great manuscript. 

Author Response

Thank you very much for your kind words and positive feedback. We greatly appreciate your thoughtful comments and are glad to hear that the revisions have addressed your concerns. Your support and constructive input have been invaluable in improving the manuscript.